# Quinolone and Organophosphorus Insecticide Residues in Bivalves and Their Associated Risks in Taiwan

**DOI:** 10.3390/molecules25163636

**Published:** 2020-08-10

**Authors:** Ching-Feng Wu, Ching-Hung Chen, Ching-Yang Wu, Chen-Si Lin, Yao-Chi Su, Ching-Fen Wu, Hsiao-Pei Tsai, Pei-Shan Fan, Chang-Hui Yeh, Wei-Cheng Yang, Geng-Ruei Chang

**Affiliations:** 1Division of Thoracic and Cardiovascular Surgery, Department of Surgery, Chang Gung University, Chang Gung Memorial Hospital, Linkou, 5 Fuxing Street., Guishan District, Taoyuan 33305, Taiwan; maple.bt88@gmail.com (C.-F.W.); wu.chingyang@gmail.com (C.-Y.W.); 2Department of Anesthesiology, Show Chwan Memorial Hospital, 1 Section, 542 Chung-Shan Road, Changhua 50008, Taiwan; consideration00@gmail.com; 3Department of Veterinary Medicine, School of Veterinary Medicine, National Taiwan University, 4 Section. 1 Roosevelt Road, Taipei 10617, Taiwan; cslin100@ntu.edu.tw; 4Department of Veterinary Medicine, National Chiayi University, 580 Xinmin Road, Chiayi 60054, Taiwan; shu@mail.ncyu.edu.tw (Y.-C.S.); cfwu@mail.ncyu.edu.tw (C.-F.W.); babybelle349@gmail.com (P.-S.F.); yehvet99@gmail.com (C.-H.Y.); 5College of Veterinary Medicine, Veterinary Teaching Hospital, National Chiayi University, 580 Xinmin Road, Chiayi 60054, Taiwan; tsaibelle@mail.ncyu.edu.tw; 6Ph.D. Program of Agriculture Science, National Chiayi University, 300 Syuefu Road, Chiayi 60004, Taiwan

**Keywords:** quinolone, organophosphorus pesticide, residues, bivalves, risk assessment

## Abstract

Bivalves, such as freshwater clams (*Corbicula fluminea*) and hard clams (*Meretrix lusoria*), are the most extensive and widely grown shellfish in land-based ponds in Taiwan. However, few studies have examined the contamination of bivalves by quinolone and organophosphorus insecticides. Thus, we adapted an established procedure to analyze 8 quinolones and 12 organophosphorus insecticides using liquid and gas chromatography–tandem mass spectrometry. Surveys in Taiwan have not noted high residual levels of these chemicals in bivalve tissues. A total of 58 samples of freshwater or hard clams were obtained from Taiwanese aquafarms. We identified 0.03 mg/kg of enrofloxacin in one freshwater clam, 0.024 mg/kg of flumequine in one freshwater clam, 0.02 mg/kg of flumequine in one hard clam, 0.05 mg/kg of chlorpyrifos in one freshwater clam, 0.03 mg/kg of chlorpyrifos in one hard clam, and 0.02 mg/kg of trichlorfon in one hard clam. The results indicated that 5.17% of the samples had quinolone insecticide residues and 5.17% had organophosphorus residues. However, the estimated daily intake (EDI)/acceptable daily intake quotient (ADI) indicated no significant risk and no immediate health risk from the consumption of bivalves. These results provide a reference for the food-safety screening of veterinary drugs and pesticides in aquatic animals. Aquatic products should be frequently screened for residues of prohibited chemicals to safeguard human health.

## 1. Introduction

Seafood aquaculture constitutes an important source of livelihood for people in several countries, and the Joint Food and Agriculture Organization of the United Nations (FAO) estimated that global aquaculture production will reach 80,000,000 t by 2050, [1]. In Taiwan, aquaculture has been developing for more than 300 years. From 1960 to the 1990s, its unique characteristics led to it becoming a rapidly intensified, diversified, and technologically intensive industry [1,2]. From 1990 to 2000, Taiwan’s aquaculture industry underwent a transition, and saltwater aquaculture was substituted for freshwater aquaculture [3]. After 2000, changes in consumption patterns and competition from global trade have forced Taiwan’s government to implement safety management policies for aquaculture, with a focus on food sanitation [4]. Despite limited land use, Taiwan’s aquaculture industry has produced world-class water fisheries and seafood producers [5,6,7]. Considering that Taiwan has few inherent economic advantages, this performance in the area of marine production is remarkable. Taiwan’s current farming methods can be categorized into sea and land ranches [4,8,9]. Taiwan’s aquaculture farms contain a wide variety of species of fish, shellfish, mollusk, crustaceans, and aquatic plants [1,7,10]. Currently, approximately 50 varieties of major and candidate species are farmed [11]. These aquatic species include crustaceans, bivalves, fishes, reptiles, gastropods, seaweeds, and amphibians. Moreover, the annual economic turnover of aquaculture during the 2010s was more than USD 10 billion, and overall production was approximately 40,000,000 t. The combined value of the cultures of gastropods and bivalves is considerable because Taiwan’s government has strongly supported the development of these aquacultures since the 1990s [1,3]. From 1973 (the onset of bivalve farming) until 2018, the average annual production of freshwater bivalves in Taiwan had reached 54,000 t and a value of more than USD 0.13 billion [11].

Aquaculture is one of the fastest growing animal food production through worldwide in recent centuries [5]. Besides their good taste, there is an upsurge in the worldwide need for essential proteins, which has required intense expansion of aquaculture, along with the use of antibiotics being increasingly to treat aquatic animal diseases. Moreover, the addition of antibiotics into animal feed results in feed conversion rates and improved growth in food-producing animals. The feeding model revealed that antibiotics are supplemented into animal food in the aquaculture industry [2,12]. Highly extensive and large-scale culture habitually cultures by Taiwan’s aquaculture farmers because the inner land is limited to use. Otherwise, their culture areas are generally located near industrial or agricultural zones, which makes it difficult to prevent microbial infection [9]. The cultured aquatic animals become more vulnerable to infections from microbes such as viral, fungal, bacterial, and parasites, a condition that necessitates the use of different veterinary drugs to prevent and treat these pathogenic infections. Excessive use of animal drugs may cause drug-resistant bacteria in food-producing animals [12]. In addition, the use of pesticides in marine farming is common practice to treat parasite infection; in particular, the aquaculture in subtropical locations and the temperate climate of Taiwan is susceptible to parasitic diseases. However, organophosphorus insecticides are banned for treating ectoparasitic or endoparasitic infection in aquatic animals [2,9]. Due to their bioaccumulation and detrimental effects on aquatic animals and humans, these chemical residues in aquatic products are essential to elucidate the level of their contamination in aquatic foods. In the last five years, investigations and studies on the levels of sulfonamides, chloramphenicol, malachite green, leucomalachite green, nitrofuran metabolites, quinolone, herbicides, organophosphorus insecticides residues have increased in aquatic animals on consumers in Taiwan [1,2,3,9,10].

The freshwater clam (*Corbicula fluminea*) and hard clam (*Meretrix lusoria*) aquaculture industry has registered the most common and novel aquaculture in ponds that are inner land-based, in Taiwan [3]. The major cultural sites of bivalves are distributed in the inner regions and Taiwanese bivalve-farming methods often adopt mixed breeding with other aquatic animals or polyculture involving waterfowls. Chemical pollution can easily accumulate through preventing or treating diseases in non-targeted shellfish. They have the potential to gradually accumulate in the edible body and cause certain organ or system damage. Because veterinary drugs and organophosphorus pesticide residues in seafood are an important public health concern, they are considered with care by the illegal/banned chemicals administration. Otherwise, the Taiwan Food and Drug Administration (TFDA) has not yet established the maximum residue limits (MRLs) for quinolones or organophosphorus insecticides in bivalves [2,9,12]. Moreover, limited reports of their residual studies did include freshwater aquaculture bivalves, especially the hard clams and freshwater clams. The chemical risk assessment of the whole range of freshwater aquaculture marine life from farms to tables should be followed in Taiwan. Therefore, this study examined quinolones and organophosphorus insecticide residues in freshwater aquaculture bivalves in main production districts in Taiwan. The estimated daily intake (EDI) of quinolones and pesticides was also examined depending on the levels of contamination in the bivalve samples. Moreover, the surveyed data are useful in evaluating the safety of consuming these shellfishes and provide their health influence as a reference for Taiwan’s government regulation.

## 2. Results

### 2.1. Levels and Ratios of Banned Quinolones in Bivalve Samples

In total, 58 bivalve samples were examined. Enrofloxacin (0.03 mg/kg) and flumequine (0.024 mk/kg) were detected separately in one freshwater clam sample (Table 1). In addition, 0.02 mg/kg of flumequine was detected in one hard clam sample. The percentage of the most common banned quinolone, flumequine, was 3.57% in freshwater clams (1/28; level: 0.024 mg/kg) and 3.33% in hard clams (1/30; level: 0.02 mg/kg). No quinolone residue was detected in 26 freshwater clam samples or 29 hard clam samples. Quinolone residue was detected at an average level of 0.0011:0.0009 mg/kg in all freshwater clams and 0.0007 mg/kg in all hard clams, respectively. Overall, quinolones (such as enrofloxacin and flumequine) were detected in only 3 (5.17%) of the 58 samples.

### 2.2. Levels and Ratios of Banned Organophosphorus Insecticidse in Bivalve Samples

The detected residual levels of prohibited organophosphorus pesticides in different bivalve samples are presented in Table 2. The percentage of the most common banned organophosphorus insecticide, chlorpyrifos, was 3.57% in freshwater clams (1/28; concentration: 0.05 mg/kg) and 3.33% in hard clams (1/30; concentration: 0.03 mg/kg). In addition, 0.02 mg/kg of trichlorfon was detected in one hard clam at a ratio of 3.33%. No residual levels of organophosphorus insecticides were detected in 27 freshwater clam and 28 hard clam samples. Chlorpyrifos residues were detected at an average level of 0.0018 and 0.0010 in all freshwater clam and hard clam samples, respectively. The average level of trichlorfon detected was 0.0007 mg/kg in hard clam samples. Overall, residues of organophosphorus insecticides (such as chlorpyrifos and trichlorfon) were detected in only 3 (5.17%) of the 58 samples.

### 2.3. EDI of Quinolone Residues in Bivalves Consumed by Taiwanese Adults

The EDIs estimated from the average residual concentrations of enrofloxacin in Taiwanese men and women were 0.808 and 0.618 ng/kg body weight/day, respectively (Table 3). Additionally, the EDIs of flumequine in Taiwanese men and women were 1.292 and 0.989 ng/kg body weight/day, respectively. The ADIs of enrofloxacin and flumequine residues in food were 0.002 and 0.03 mg/kg, respectively, as stipulated by the FAO/World Health Organization Expert Committee on Food Additives (JECFA) [12,13]. The estimated EDI values were less than the JECFA-recommended ADI levels of enrofloxacin and flumequine. For Taiwanese men and women, the percentages of EDI/ADI in enrofloxacin were 0.04% and 0.031%, respectively, and those in flumequine were 0.004% and 0.003%, respectively. Overall, the risk of consuming excessive quinolones from bivalves is low: the EDI levels were less than 0.1% of the ADI for both Taiwanese men and women.

### 2.4. EDI Levels for Residues of Organophosphorus Insecticides Ingested in the Consumption of Bivalves by Taiwanese Adults

The EDIs determined from the mean residual levels of chlorpyrifos in Taiwanese men and women were 2.261 and 1.731 ng/kg body weight/day, respectively (Table 4). Additionally, the EDIs of trichlorfon in men and women were 0.485 and 0.371 ng/kg body weight/day, respectively. The estimated levels of EDI were less than the JECFA-recommended ADI of 0.01 mg/kg for chlorpyrifos and 0.02 mg/kg for trichlorfon, respectively [2,9]. In addition, the percentages of EDI/ADI for chlorpyrifos were 0.023% and 0.017% for Taiwanese men and women, respectively. The EDI levels, estimated as percentages of ADI, for trichlorfon were 0.003% and 0.002% for Taiwanese men and women, respectively. Overall, the risk of consuming excessive organophosphorus insecticides from bivalves was low: the EDIs were less than 0.1% of the ADI for both Taiwanese men and women.

## 3. Discussion

A total of 58 samples of aquaculture bivalves were collected from different regions in Taiwan for a residue analysis of eight quinolones (ciprofloxacin, danofloxacin, difloxacin, enrofloxacin, fleroxacin, flumequine, marbofloxacin, and sarafloxacin) and 12 organophosphorus insecticides (profenophos, chlorpyrifos, methamidophos, diazinon, trichlorfon, fenamiphos, fenthion, fenitrothion, malathion, formothion, phoxim, and triazophos). As stipulated by the TFDA, residues of quinolones and organophosphorus insecticides must not be present in bivalve products, and the use of quinolones and organophosphorus insecticides in bivalve farms is illegal. The bivalves were thus screened for residues of these chemicals. The detected levels of residues indicated that food safety standards were being adhered to.

The detection limit of quantification (LOQ) for all quinolones in bivalve samples using liquid chromatography–tandem mass spectrometry (LC–MS/MS) was 5 ng/g (Appendix A). The identified levels satisfied the TFDA-recommended LOQs for quinolone contamination in edible aquacultured foods, chicken, milk, and livestock [14] as well as the legal prohibition against quinolone residue in animal viscera [15]. However, no reference LOQs for residues of organophosphorus pesticides in aquatic foods have been made available by the TFDA. Nevertheless, the TFDA-recommended LOQ for detected organophosphorus insecticide residues is 10 ng/g in pork, beef, and chicken muscles [16]. Applications of earlier methods developed to identify organophosphorus pesticide residues in shellfishes, cephalopods, crustaceans, and fish have employed a diverse range of equipment. For example, a gas chromatograph (GC) with a nitrogen phosphorus detector revealed the detected limits of chlorpyrifos diazinon, profenophos, and fenitrothion in green mussel to be 0.5 ng/g [17]; gas chromatography–mass spectrometry revealed the detected limits of diazinon, fenthion, disulfoton, malathion, propetamphos, and triazophos in fish to be 7–15.2 ng/g [18]; LC–MS/MS coupled gel filtration chromatography revealed the detected limits of chlorpyrifos, profenofos, trichlorphon, phosmet, triazophos, malathion, and dimethoate in shrimp to be 0.05–0.2 ng/g [19]; and LC–MS/MS revealed the detected limits of fenamiphos, fenthion, methamidophos, and profenophos in shrimp to be 5 ng/g. Similarly, GC–MS/MS revealed the detected limits of chlorfenvinphos, chlorpyrifos, diazinon, and fenitrothion in shrimp to also be 5 ng/g [9]. In comparison to the LOQs of these studies, our analytical procedure employed lower LOQs (all at 5 ng/g) for the determination of organophosphorus pesticide multiresidues (Appendix A), which proved to be more useful in identifying traces of organophosphorus pesticide. Thus, the analytical procedures described in this study can be employed to detect incidents of quinolone contamination or organophosphorus pesticide contamination in bivalves.

The present study employed the TFDA-recommended analytical method for detecting trace quinolone residues [14] as well as a method for detecting traces of organophosphorus insecticide developed by the European Committee for Standardization, known as the QuEChERS approach [2,9]. To validate aquatic samples for chemical analysis, the TFDA [5,18] proposed a recovery rate that is acceptable at 70–120% with relative standard deviation (RSD) < 20%, acceptable at 60–125% with RSD < 30%, and acceptable at 50–125% with RSD < 35% for chemical residues in food matrices that are identified in the 0.01–0.1, 0.001–0.01, and < 0.001 mg/kg ranges, respectively. In this study’s results, residues of quinolones and organophosphorus insecticides that were detected in the range of 0.01–0.1 mg/kg exhibited a recovery rate of 96.35–100.76% with RSD < 12% (Appendix A) and 95.47–114.53% with RSD < 13% (Appendix A), respectively. In addition, the amounts of spiked analytes employed at lower and higher levels were 5 and 25 ng/g, respectively for the eight quinolones and 12 organophosphorus insecticides. Therefore, all quinolone and organophosphorus pesticide concentrations were within the accepted range [2,12,20]. The proposed method was therefore validated, and it could be applied to the detection of quinolone and organophosphorus insecticide residues in the bivalve samples.

The LC–MS/MS revealed the presence of quinolones in only 5.17% of the 58 bivalve samples, with positive indications of the TFDA-banned quinolones of enrofloxacin and flumequine. Few surveys have been conducted to detect quinolones in bivalves. Nevertheless, quinolone residues of enrofloxacin were identified in 40.0% of 21 samples of aquaculture fish obtained in the period 2009–2011 in China [21], in 6.0% of 100 samples of fish and shrimp samples obtained during 2011 in Vietnam [22], in 16.67% of 30 fish and shrimp samples obtained during the period 1994–2004 in a Canadian Total Diet Study [23], and in 2.70% of 37 fish and shrimp samples obtained during 2010 in Japan [24]. The surveys conducted by TFDA in Taiwan during the period 2013–2016 for quinolone traces in aquacultured foods identified no unlawful use of quinolones as antibiotic agents in bivalves [25,26,27,28,29]: the rate of identification of prohibited quinolone residues in freshwater and hard clam samples was 0% in 2012 (18 samples) [25], 2013 (36 samples) [26], 2014 (34 samples) [27], 2015 (20 samples) [28], and 2016 (10 samples) [29]. In comparison with the 2012 TFDA report, the violation ratio for quinolone residues in 25 shrimp samples was 4.0% [25]. Moreover, the violation ratio of quinolone residues in Taiwanese soft-shell turtle samples was 20.0% in 10 samples in 2014 [27], 13.33% in 15 samples in 2015 [28], and 3.12% in 32 samples in 2016 [29]. Our present findings are inconsistent with the surveys conducted by the TFDA, which is attributable to differences in sample size or sample species. Moreover, the samples in the current study were obtained from aquaculture production locations in Taiwan, whereas imported bivalves may have formed part of the TFDA-conducted surveys. Additionally, our study employed a larger bivalve sample size and analyzed more quinolones. Therefore, several categories of banned quinolone residues for bivalves in Taiwan were appropriately estimated in the present study.

The major quinolone residue was enrofloxacin at the highest level of 0.03 mg/kg. This is consistent with results for whiteleg shrimp [12], soft-shell turtle [27], and yellow croaker in Taiwan [30] and those for catfish in Vietnam [31]: enrofloxacin is a quinolone frequently applied in common tropical aquatic cultures to treat and prevent bacterial infestations. Moreover, in all samples of bivalves, flumequine was the main residue at 3.45% (2/58), followed by enrofloxacin at 1.72% (1/58). The present results are comparable to those described in the TFDA survey. Compared with the TFDA findings, the higher violation ratio of quinolone residues with flumequine in shrimp samples was 4.0% in 2012 [25], and those in soft-shell turtle samples were 6.67% in 2015 [28] and 3.13% in 2016 [29]. Based on the information available, we conclude that flumequine should continue to be utilized as an antibacterial agent in aquatic products for its effectiveness and affordability. Other surveys in Asia [32], North Africa, the Mediterranean, and the Atlantic [33,34] have reported that flumequine is the most commonly employed second-generation antibiotic in aquaculture, by virtue of its better stability in resisting degradation by bacteria in an aquatic environment. Moreover, TFDA studies have positively identified ciprofloxacin in soft-shell turtle samples [27,28] and in shrimp samples [35]. These findings confirm that Taiwanese aquaculture regularly uses quinolones in edible aquatic animals. Thus, combining our findings with the findings derived from other surveys illustrates the exposure of Taiwanese people to trace quinolone concentrations through the consumption of bivalves. Regulatory authorities and producers in Taiwan must therefore continually screen seafood and avoid sources of chemical pollution to ensure the safety of commercially available foods.

Regarding the analysis of 12 organophosphorus insecticides, the bivalve samples tested positive for chlorpyrifos and trichlorfon. Chlorpyrifos was the most frequently detected residual organophosphorus chemical. In all bivalve samples, the maximum detection rate for chlorpyrifos was 3.45% (2/58), in contrast to the 1.72% (1/58) for trichlorfon. Several aquaculture farms in Taiwan use organophosphorus pesticides to treat and control ectoparasite infections in the culture of grass shrimp and fish [36,37]. In addition, the use of trichlorfon is permitted in Taiwan for the treatment of ectoparasites in fish, such as *Anguilla* spp., Ostariophysi, Perciformes, and Teleostei. However, the detection of organophosphorus insecticide residue in the bivalves suggests that these insecticides have been illegally used either to directly treat parasitic infections in bivalves or to control the parasitic diseases in other aquatic animals that are caused by mixed breeding in aquatic ponds. Although no official survey has been conducted on the occurrence of organophosphorus pesticides in aquatic animals in Taiwan, these residues were detected in an earlier study conducted by the TFDA on organochlorine pesticides in shellfish and fish [38]. However, organophosphorus pesticides are the most extensively utilized pesticide worldwide and are potent inhibitors of butyrylcholinesterase; thus, they greatly and negatively affect aquatic as well as terrestrial vertebrates [2,9,39]. A survey conducted by Sun et al. [40] reported a detection rate of 11.37% for organophosphorus insecticides (in 607 fish samples) in Taiwan during the period 2001–2003, where they did not sort their sample into whether the fish were sourced from fish markets, fish farms, or traditional and regional supermarkets. Sun et al. also reported a detection rate of 16.83% for organophosphorus insecticides in 814 fish samples that were sourced during the period 2002–2004 from Taiwan markets [41]. The measurements in our findings were significantly less than those reported by Sun et al. due to differences in sample size, the variety of organophosphorus insecticide categories, and differences in collection sources [40,41]. Moreover, a reduction in the use of the veterinary drug as a result of the Taiwanese government’s enactment of the national action plan since 2006 [39] may also have resulted in fewer detected violations of laws against chemical residues in seafood.

Chlorpyrifos was the most common organophosphorus insecticide detected in the bivalve samples of the present study. This differs from the findings of a comparable bivalve study in Indonesia, where levels of fenitrothion residue were highest [17]. However, our findings are consistent with another aquatic survey on organophosphorus insecticide residues detected in fish in Egypt [42] and shrimp in Taiwan [9]. Although chlorpyrifos degrades quickly in the environment, these studies have found that widespread chlorpyrifos usage in the aquaculture industry has increased overall toxicity to aquatic animals [2]. Chlorpyrifos remains a commonly used organophosphorus pesticide for killing pests in feed additives in agriculture industries [43]. Due to its neurotoxic effects and a decrease in fetal birth weight caused by organophosphorus insecticides, Taiwan’s government has prohibited the use of organophosphorus pesticides in aquaculture and has set strict MRLs for animal husbandry [9]. In addition, we identified chlorpyrifos levels of 0.0014 mg/kg in bivalves, considerably lower than the level of 0.463 mg/kg detected in Taiwanese farmed fish between 2002 and 2004 [41]. These surveys have also indicated that chlorpyrifos is detected more frequently in aquaculture products in Taiwan, which is linked to organophosphorus insecticide being readily available in pesticide stores due to the loose control of the substance by Taiwan’s government [40,41]. In addition, chlorpyrifos was listed fourth in terms of usage among all pesticides, and its production reached a total of 1659 t between 2012 and 2016 [44]. The sources of chlorpyrifos contamination could be aquaculture feed and the aquatic environment, such as sediments and water. However, information on chlorpyrifos contamination in bivalve feed is scant. We therefore suggest for more studies to be conducted in the detection of chlorpyrifos content in bivalves or other aquatic feeds. We recommend doing so because organophosphorus insecticide can be added to treat ectoparasite infections in other mixed breeding aquatic animals, although this practice is not widely adopted.

The TFDA has currently not defined any recommended MRL levels for quinolones or organophosphorus insecticides in bivalves [9,12,15]. In the present study, all samples with quinolone or organophosphorus insecticide levels lower than 1 mg/kg were in violation of Taiwanese regulations. In Taiwan, positive bivalve samples do not need to be taken off the market, even if they have trace residues. To examine the risk of contamination by chemical residues when consuming bivalves, several parameters can be adapted in guidelines to reduce the risk of exposure to food contaminants; these parameters include ADI, tolerable daily intake (TDI), target hazard quotient, and total target hazard quotient [2,9,12]. Proposed by the JECFA and US Environmental Protection Agency, the EDI is a novel and rigorous approach for more precisely determining chronic dietary consumption [2,12]. The absence of an observed effect is defined as EDI/ADI < 1%, where the ADI values are provided by the JECFA [45]. In the present study, all EDI/ADI values were within their corresponding limits. Because few samples tested positive for quinolone residue, the EDI levels indicated a lower dietary intake of quinolones in Taiwanese bivalve consumption relative to that indicated by the corresponding ADIs of enrofloxacin and flumequine set by the FAO/WHO. In the present study, because the EDI/ADI values of quinolone residues in bivalve samples were lower than 0.1%, the risk was assessed to be negligible [1,2,12]. Furthermore, in the population of Taiwan, aquacultured bivalve intake resulted in exposure to much lower levels of chlorpyrifos and trichlorfon; this was because their EDIs were considerably lower than their corresponding ADIs. Our findings are similar to previous findings on fish consumption [10,12,46]; these findings indicate a negligible risk of quinolone consumption because EDI/ADI < 0.1%. Therefore, residual quinolone and residual organophosphorus insecticide in aquacultured bivalves in Taiwan are, at these levels, unlikely to adversely affect the health of the Taiwanese population.

Overall, in the bivalve samples of our study, the detected levels and ratios of quinolone and organophosphorus pesticide residues remain low and within permissible limits; thus, the presence of these substances do not constitute harmful pollution. However, in a few cases, non-negligible residues of quinolone and organophosphorus insecticide were detected in shellfish. We provided these analytical results to the Taiwan government to inform them that because measurements were made on the samples while they were on the farm, the samples in violation of regulations were blocked from getting into the food supply. Crucially, these prosocial practices are adopted in Taiwan because the penalty for rule breaking has no maximum, as stipulated in the Taiwanese Act Governing Food Safety and Sanitation. However, the (albeit slight) presence of detected residues entails a need for sustained aquaculture monitoring. With regard to policy making, we advise Taiwan’s regulatory authorities to frequently screen aquatic products and survey possible sources of pollution to ensure consumer food safety. Furthermore, because antibiotics and pesticides may have unwanted effects on human and animal health, parallel studies examining the effects of such contamination are crucial.

## 4. Materials and Methods

### 4.1. Samples, Chemicals, and Reagents

Altogether, 58 bivalve samples (28 freshwater clam, and 30 hard clam samples) were obtained from Taiwanese aquafarms in predominant areas of production situated in Changhua, Yunlin, Chiayi, Tainan, Kaohsiung, and Hwalien during June 2018 to December 2019. According to the survey in 2018 conducted by Fisheries Agency of Taiwan, these bivalves were grown on a large scale in Taiwan [8]. From the collected bivalve samples (each sample at ~600 g), each bivalve mussel was removed and cleaned, followed by homogenization and storage at −20 °C till further analysis. In these measurements, from each composite sample we took three replicates, and the mean concentration of each selected chemical was determined.

Ciprofloxacin, enrofloxacin, difloxacin, danofloxacin, flumequine, fleroxacin, marbofloxacin, and sarafloxacin were procured from Sigma–Aldrich (St. Louis, MO, USA) (all > 95.0% purity). In addition, the organophosphorus insecticide related chemicals, such as phoxim, chlorpyrifos, fenitrothion, diazinon, malathion, fenthion, methamidophos, trichlorfon and profenophos, (all > 97.0% purity) were from Dr. Ehrenstorfer GmbH (Augsburg, Germany). Additionally, fenamiphos (99.5% purity) was bought from Bayer CropScience AG (Monheim, Germany), formothion (96.0% purity) was supplied by Sandoz India (Mumbai, India), and triazophos (99.5% purity) was purchased from ChemService (West Chester, PA, USA). Merck (Darmstadt, Germany) provided chromatography-grade chemicals including formic acid (FA), acetone, methanol (MeOH), n-pentane, n-hexane, ethyl acetate, anhydrous sodium sulfate (ASS), and acetonitrile (ACN). We also procured 15-mL QuEChERS cleanup tubes (Agilent SampliQ QuEChERS EN fatty dispersive-SPE kit, p/n 5982-5156) and a QuEChERS extraction salt packet (Agilent SampliQ QuEChERS EN Extraction kit, p/n 5982–5650; mixture constituents: 1-g sodium citrate, 1-g NaCl, 0.5-g citric acid disodium salt, and 4-g anhydrous magnesium sulfate) from Agilent Technologies (Wilmington, DE, USA).

### 4.2. Instruments and Apparatus

A vortex (type 37600 mixer, Thermolyne, Dubuque, IA, USA), a N-Evap-111 nitrogen evaporator (Organomation Associates Inc., Berlin, MA, USA), a Allegra X-22R centrifuge (Beckman Coulter Inc., Fullerton, CA, USA), and a nitrogen generator (Model 05B, System Instruments Co., Tokyo, Japan) were utilized to prepare the samples.

For LC-MS/MS analysis quinolones, a mass spectrometer (ABI 4000 QTRAP, Applied Biosystems, Foster City, CA, USA) in an electrospray ionization (ESI) mode and an UltiMate 3000 HPLC system (Thermo Fisher Scientific, Waltham, MA, USA) were employed in this study. HPLC separation was were carried out on an Acquity HSS T3-column (2.1 mm × 1.8 μm, 100 mm; Waters, Milford, MA, USA). For detecting the residues of LC-amenable organophosphorus insecticides, the LC/MS–MS equipment consisted an UltiMate 3000 HPLC system and a mass spectrometer (TSQ quantiva triple quadrupole; Thermo Fisher Scientific, Austin, TX USA). Furthermore, LC-amenable organophosphorus insecticides were separated using a Waters CORTECS UPLC C18 column (2.1 mm × 1.6 μm, 100 mm). For determining GC-amenable organophosphorus insecticides, GC–MS/MS was conducted on a GC system (Thermo Scientific Trace 1310; Thermo Fisher Scientific, Austin, TX, USA) and a mass spectrometer (TSQ 8000 triple quadrupole, Thermo Fisher Scientific) that was coupled with Rxi^®^-5Sil MS column (fused silica) (0.25 mm × 0.25 μm, 30 m; Restek, Bellefonte, PA, USA).

### 4.3. Standard Solution Preparation

Stock solutions of individual quinolones and organophosphorus insecticide standards were prepared as follows: 100 mg of each analyte was precisely weighed and dissolved in 100 mL of MeOH (ciprofloxacin, enrofloxacin, difloxacin, danofloxacin, flumequine, fleroxacin, marbofloxacin, and sarafloxacin), ACN (fenamiphos, fenthion, methamidophos, phoxim, profenophos, and trichlorfon), or acetone (chlorpyrifos, diazinon, fenitrothion, formothion, malathion, and triazophos) according to analyte solubility. To prepare the working standard mixtures, all stock solution types were mixed and diluted to 1 mg/L. All solutions were stored at −20 °C and then equilibrated to room temperature prior to use. The working standard solutions were then used to derive a series of calibration standards through serial dilution within the range of 1–500 ng/mL.

### 4.4. Sample Preparation and Extraction

For detecting quinolones residues, we employed TFDA’s guidelines for multiresidue analysis of residues of veterinary drugs in foods, that needed the cleaning and homogenization of bivalve samples initially [14]. Briefly, 5 g of the homogenate and 25 mL of ACN in 5% MeOH were mixed on a vortex mixer for 3 min. Then, the homogenate was added with 10 g of ASS and mixed for 10 min; afterwards, centrifugation was done at 3500× *g* at 4 °C for 10 min, and the supernatant was stored. The remaining tissue pellets were re-extracted with 25 mL of ACN in 5% MeOH, followed by re-centrifugation. The supernatant was mixed with the earlier separated ACN layer. The resulting mixture was taken for liquid–liquid extraction in a separating funnel. The filtrate was then mixed with 30 mL of ACN-saturated n-hexane and thoroughly mixed using a vortex mixer for 10 min. The ACN-extracted layer was separated and dried at 40 °C under nitrogen. The residue obtained was dissolved in 1 mL of 50% MeOH and filtered using a 0.2-µm polyvinylidene fluoride (PVDF) filter (Whatman, Maidstone, UK). The resulting filtrate was transferred to an autosampler vial for subsequent injection into a chromatograph.

Insecticide residues in bivalve samples were measured using the QuEChERS extraction procedure developed by the European Committee for Standardization [2,10,47]. First, 10 mL of ACN and 10 g of a homogenized bivalve sample were vigorously mixed in a 50-mL centrifuge tube for 1 min; QuEChERS extraction salt was then added and mixed on a vortex for 1 min. The mixture was then centrifuged for 5-min at 3000× *g*. The crude ACN extract (≈6 mL) was then placed into QuEChERS cleanup tubes. The ACN layer was vigorously mixed for 2 min and centrifuged for 5 min at 3000× *g*. 1 mL of the extract was then filtered through a 0.2-μm PVDF membrane and transferred into an LC/MS–MS autosampler vial. Another 1 mL of the extract was dried almost completely under nitrogen at 40 °C. The resultant residue was first dissolved in a 1 mL mixture of 1:1 (*v*/*v*) n-hexane and acetone [2,9,47], then filtered through a 0.2-μm PVDF filter, and finally placed in a GC–MS/MS autosampler vial.

### 4.5. LC–MS/MS Parameters

The injection volume for detecting the quinolone and organophosphorus insecticide residues was kept at 10 µL. The binary mobile phase comprised of eluents A (0.1% FA) and B (0.1% FA in MeOH), and the gradient of the mobile phase was developed as follows: 5% eluent B from 0 to 2 min (flow rate, 0.3 mL/min); followed by a step increase of eluent B from 2 to 3 min to 20%, from 3 to 6 min to 25%, from 6 to 8.5 min to 30% and to 40% from 8.5 to 15.0 min; then it was linearly increased to 100% from 15 to 16.5 min; and finally by 18 min, eluent B was decreased to 4% and maintained between 18 to 20 min. MS was performed in positive ESI mode by monitoring the two most abundant MS/MS (precursor/product) ion transitions for each analyte by employing an MRM program. The MS parameter settings are as follows: collision gas argon pressure, 0.12 mL/min; desolvation flow, 1000 L/h; source temperature, 150 °C; desolvation temperature, 500 °C; dwell time for every MRM transition, 5 ms; cone gas flow, 50 L/h; capillary voltage, 3 kV. The precursor and corresponding product ions with optimum collision energy obtained through the MRM detection for 8 quinolones and LC-amenable 6 organophosphorus insecticides, are provided in Appendix A, respectively.

### 4.6. GC–MS/MS Parameters

GC–MS/MS analysis was conducted in positive and negative electron-impact ionization interface modes. The carrier gas, helium, was pumped at a constant flow rate of 1 mL/min. The injector temperature was 280 °C. Moreover, the oven temperature was set at 60 °C—it was maintained initially isothermal for 1 min, next raised to 150 °C at 40 °C/min, and finally kept at 300 °C for 8 min. The set source and transfer-line temperatures were 300 and 250 °C, respectively. In the splitless mode, the injection volume was determined to be 10.0 μL. In the collision chamber (second quadrupole), these ions were collision-activated with argon at 4.4 mTorr. The precursor and corresponding product ions with optimum collision energy obtained through the MRM detection for GC-amenable 6 organophosphorus insecticides are listed in Appendix A.

### 4.7. Quality Assurance and Validation

To authenticate our method, we determined its recovery, repeatability, linearity, and LOQ [2,7,10]. To evaluate the method’s repeatability and recovery, we spiked blank samples (in triplicate) with a standard analyte mixture at 5 (low) and 25 (high) ng/g concentrations to analyze quinolones and LC- and GC-amenable organophosphorus insecticides. Recovery was calculated by comparing the observed concentrations of samples spiked before extraction with blanks spiked at the same concentration following extraction.

In this study, the repeatability results are presented in terms of the RSD (%); in addition, the LOQs were defined as the analyte concentration that generated a chromatogram peak signal that was 3–10 times greater than the background noise. To ascertain linearity, matrix-matched calibration was performed utilizing blank sample extracts followed by the addition of the corresponding amount of the working target compound solution at concentrations of 1–500 ng/mL. The linearity of the calibration curves was determined using least-squares linear regression (R^2^ ≥ 0.990) in the examined concentration range. All sample concentrations lower than their corresponding LOQs were considered to be undetectable [1,12,47].

### 4.8. Quinolones and Organophosphorus Insecticide EDIs Compared with ADIs

To evaluate the extent of exposure to veterinary drug residues in Taiwanese people, through bivalves, the related EDIs of the quinolones and organophosphorus insecticide residues were measured. The JECFA-recommended ADIs for these residues were employed for comparison. EDI calculation was conducted as follows [2,7,9]: EDI (ng/kg/day) = [(daily fish consumption in g/day) × (mean veterinary drug concentration in ng/g)]/(human body weight in kg)

In addition, data about Taiwan residents’ daily seafood intake were retrieved from the results of the National Nutrition and Health Survey by Ministry of Health and Welfare, Taiwan: 96.9 g for men and 74.2 g for women [48]. The mean Taiwanese body weight was considered to be 60 kg [9,12,48]. The maximal EDIs were obtained from maximum residue concentrations.

## 5. Conclusions

We formulated a sensitive and efficient LC and GC–MS/MS-based technique for identifying quinolone and organophosphorus insecticide residues in bivalves. The most frequently detected quinolone residues in all bivalve samples were enrofloxacin and flumequine; the most frequently detected organophosphorus insecticide residues were chlorpyrifos and trichlorfon. However, all the chemicals were detected in minute amounts, well below the TFDA-recommended MRLs, in all 52 bivalve samples. This suggests that quinolone and organophosphorus insecticide usage during the large-scale breeding of bivalves in Taiwan does not result in a high level of contamination. Additionally, EDI levels for these chemicals in Taiwanese adults were noticeably less than the JECFA-defined ADI levels—indicating that no imminent health risk is posed by the consumption of aquacultured animals. Thus, low enrofloxacin, flumequine, chlorpyrifos, and trichlorfon intake through the consumption of polluted bivalves in Taiwan appears to present no significant threat to the health of Taiwanese people. Taiwanese regulatory authorities and aquafarmers may use our results to improve aquaculture-related food safety regulations.

However, and crucially, we must stress that we are not aware of toxins other than quinolones and organophosphorus insecticides that are employed in the production of bivalves; these unknown toxins may be present in the general population’s daily meat intake at hazardous levels greater than their ADIs. Therefore, it is necessary to establish a database on veterinary antibiotic and insecticide consumption through bivalves, where the constituent data can underpin an appropriate monitoring and management framework. Moreover, aquatic edible animals should be constantly monitored to identify chemical residues and guarantee food safety.

## Figures and Tables

**Table 1 molecules-25-03636-t001:** Residual detection of banned quinolones in bivalve samples obtained from June 2018 to December 2019.

Bivalve	Targets Detected	Surveyed Samples	No. of Residue	Detected Residues ^1^ (mg/kg)	Average ^2^ (mg/kg)	Residual Ration ^3^ (%)
freshwater clam	enrofloxacin	28	1	0.030 ± 0.0003	0.0011	3.57
	flumequine		1	0.024 ± 0.0002	0.0009	3.57
hard clam	flumequine	30	1	0.020 ± 0.0005	0.0007	3.33
Total	enrofloxacin	58	1	0.030	0.0005	5.17
	flumequine		2	0.020–0.024	0.0008	

^1^ Values are given in terms of mean ± SEM, ^2^ Estimated from all examined samples (i.e., detected and undetected), ^3^ Samples with residual concentrations at or below quantification/detection limits.

**Table 2 molecules-25-03636-t002:** Residual detection of banned organophosphorus insecticides in bivalve samples obtained from June 2018 to December 2019.

Bivalve	Targets Detected	Surveyed Samples	No. of Residue	Detected Residues ^1^ (mg/kg)	Average ^2^ (mg/kg)	Residual Ration ^3^ (%)
freshwater clam	chlorpyrifos	28	1	0.050 ± 0.0005	0.0018	3.57
hard clam	chlorpyrifos	30	1	0.030 ± 0.0006	0.0010	3.33
	trichlorfon		1	0.020 ± 0.0005	0.0007	3.33
Total	chlorpyrifos	58	2	0.030–0.050	0.0014	5.17
	trichlorfon		1	0.020	0.0003	

^1^ Values are given in terms of mean ± SEM, ^2^ Estimated from all the examined samples (i.e., detected and undetected), ^3^ Samples with residual concentrations at or below quantification/detection limits.

**Table 3 molecules-25-03636-t003:** Estimated daily intake of quinolone residue in bivalves by Taiwanese adults as percentage of acceptable daily intake quotient (ADI).

Quinolones	EDI (ng/kg Body Weight/Day)		EDI% of ADI		ADI (FAO/WHO) (mg/kg Body Weight/Day)
	Male	Female	Male	Female	
Enrofloxacin	0.808	0.618	0.040	0.031	0.002
Flumequine	1.292	0.989	0.004	0.003	0.030

**Table 4 molecules-25-03636-t004:** Estimated daily intake of organophosphorus insecticide residue in bivalves by Taiwanese adults as percentage of ADI.

Insecticides	EDI (ng/kg Body Weight/Day)		EDI% of ADI		ADI (JECFA) (mg/kg Body Weight/Day)
	Male	Female	Male	Female	
Chlorpyrifos	2.261	1.731	0.023	0.017	0.010
Trichlorfon	0.485	0.371	0.003	0.002	0.020

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
