# Peer review of "Quinolone and Organophosphorus Insecticide Residues in Bivalves and Their Associated Risks in Taiwan"

_molecules, 2020, doi:10.3390/molecules25163636_

Round 1
Reviewer 1 Report
Manuscript is of very poor language quality and, because of that the scientific evaluation is barely possible. It is very hard to understand what Authors have in mind due to very serious grammar and stylistic errors, especially in terms of results and discussion. My remarks are focused in the Material and method section, which was better prepared than the rest of manuscript.
This manuscript have to be firstly checked by the native speaker and an expert in the study area and then resubmitted for proper scientific evaluation.
Material and methods:
- Line 339: fish or bivalves/clams? Please be precise!
- Lines 344-351: Why these quinolones and organophosphorus compounds were chosen for detection?
- Line 379-380: Did you mix the acetonitrile and methanol solutions? What type of solvent was used for dilution of this mixture to 1mg/L?
- Line 407: Why sample residues was dissolved in the mixture of hexane and acetone, while standards were dissolved in the acetonitrile and methanol? How do you compare mass spectra obtained in very different solvents?
- Line 454: Did Thai people consumed only clams? What is the composition of eaten seafood ingredients? When the National Nutrition and Health Survey by Ministry of Health and Welfare was conducted? Have there been any changes in the pattern of seafood consumption since then?
Reviewer 2 Report
Dear Authors,
this is an interesting manuscript on the quinolones and organophosphorus insecticide residues in bivalves as a measure of risk assessment. In total the concept is interesting and important, the experiments well organized and the data well presented. I only see some minor language editing before publishing.
Reviewer 3 Report
Manuscript ID: molecules- 857982 entitled “Quinolone and organophosphorus insecticide residues in bivalves determined by liquid/gas chromatography–tandem mass spectrometry and their risk assessment” is interesting and has a potential to be helpful for researchers working in the food safety area.
The Authors should take into account the following comments:
- The research described in the paper concerns samples originating in Taiwan. In this regard, I believe that the authors should consider changing the title of the work, and add information about it.
- References cited in the paper are not always references to source materials, e.g. in lines 53-55 ref 5 and 6.
- In the reference list, authors should include relevant DOI numbers.
- All abbreviations should be explained where they appear for the first time, for example: the EDI/ADI in the Abstract.
Round 2
Reviewer 1 Report
Manuscript was improved according to the Reviewer' sugegstion and now may be accepted for publication.